# Machine Learning for Event Reconstruction in the CMS Phase-2 High Granularity Calorimeter Endcap

**Théo Cuisset on behalf of the CMS Collaboration[1]⋆,**

**1** Laboratoire Leprince-Ringuet, École Polytechnique - CNRS/IN2P3, Palaiseau, France

⋆ theo.cuisset@polytechnique.edu ,

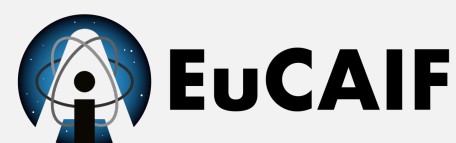

*The 2nd European AI for Fundamental Physics Conference (EuCAIFCon2025) Cagliari, Sardinia, 16-20 June 2025*

## Abstract

**The high-luminosity era of the LHC will offer greatly increased number of events for more precise Standard Model measurements and Beyond Standard Model searches, but will also pose unprecedented challenges to the detectors. To meet these challenges, the CMS detector will undergo several upgrades, including the replacement of the current endcap calorimeters with a novel High-Granularity Calorimeter (HGCAL). To make optimal use of this innovative detector, new and original algorithms are being devised. A dedicated reconstruction framework, The Iterative Clustering (TICL), is being developed within the CMS Software (CMSSW). This new framework is designed to fully exploit the high spatial resolution and precise timing information provided by HGCAL. Several key ingredients of the object reconstruction chain already rely on Machine Learning (ML) techniques and their usage is expected to further develop in the future. The existing reconstruction strategies will be presented stressing the role played by ML techniques to exploit the information provided by the detector. The areas where ML techniques are expected to play a role in the future developments will be also discussed.**

## 1 Introduction

The High-Luminosity LHC (HL-LHC) program [1] aims to increase the integrated luminosity recorded by the LHC by roughly an order of magnitude. As a result, the number of proton-proton interactions per bunch crossing (referred to as *pileup*) could rise up to 200, compared to an average of 64 in the current LHC operations. The existing electromagnetic endcap calorimeters of CMS [2,3], based on scintillating crystals, were originally designed to tolerate an accumulated dose corresponding to 500 fb$^{-1}$ of integrated luminosity. This limit will be reached by the end of Run 3 in 2026, after which the crystals would suffer a loss of transparency incompatible with the CMS physics goals. To address this issue, the CMS Collaboration has decided to replace the current endcap calorimeters with a new high-granularity sampling calorimeter,

known as the HGCAL (High Granularity Calorimeter). The HGCAL is designed to deliver excellent physics performance in the endcap regions, ensuring precise electromagnetic energy measurements (crucial for processes such as $H \rightarrow \gamma\gamma$) and accurate timing, even under the extreme radiation and pileup conditions of the HL-LHC. The unprecedented number of readout channels and high detector occupancy (illustrated in Figure 1) call for the development of new reconstruction algorithms.

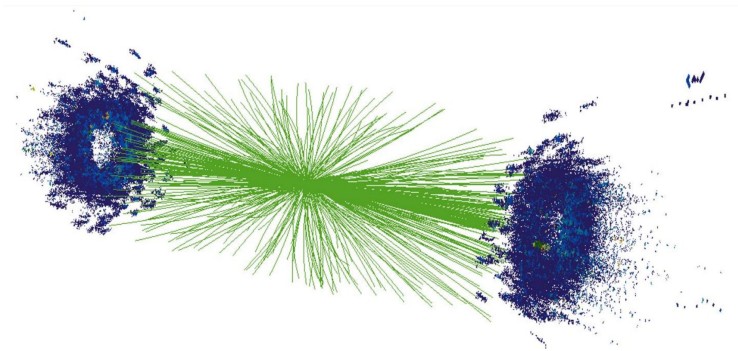

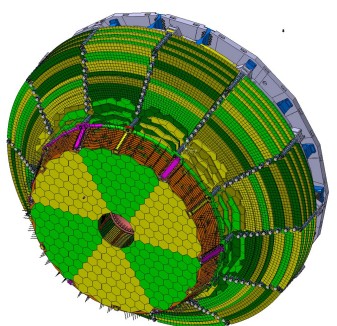

Figure 1: Simulated event display at 200 pileup, showing particle trajectories in the tracker (green) and energy deposits in the calorimeter endcaps (blue). *Source: CMS Collaboration*

Figure 2: Engineering view of one HGCAL endcap. *Source: [4]*

The HGCAL is a sampling calorimeter comprising electromagnetic (EM) and hadronic sections (Figure 2) [5]. The EM section comprises 26 layers of hexagonal silicon sensors (about 1 cm across), alternating with copper and lead absorbers. The hadronic section contains 21 layers, where the active medium consists either of silicon sensors or scintillator tiles—the latter used in regions farther from the beam pipe where radiation levels are lower. Steel plates serve as absorbers in this section.

In total, the calorimeter will feature about 6 million sensors, each capable of measuring both energy deposition and timing, with occupancy ranging from 60% in the front of the EM section down to less than 1% in the rear scintillator sensors [5]. This combination of fine granularity and precise timing represents a novel approach to calorimetry, enabling improved pileup rejection and particle identification, while preserving excellent energy resolution under HL-LHC conditions. Fully exploiting these capabilities requires the development of advanced reconstruction algorithms tailored to this new detector concept.

## 2   Clustering steps in HGCAL reconstruction

Reconstruction in the HGCAL is performed within the TICL framework [6], specifically designed to exploit its fine granularity and precise timing. For each triggered event, signals from all sensors are processed, starting at the most granular level into *rechits*, which record the position, energy, and time of individual deposits (Figure 3). To reduce computational complexity, *rechits* are first clustered within each calorimeter layer using the CLUE algorithm [7], forming two-dimensional clusters corresponding to transverse shower slices. These are then combined by the CLUE3D algorithm into *tracksters*, three-dimensional objects that represent full particle showers. Each trackster stores variables such as total energy, position, timing, and the results of a Principal Component Analysis (PCA) applied to its constituent 2D clusters, providing estimates of the shower's direction, length, and radius. Unlike the current homogeneous electromagnetic calorimeter of CMS, the fine segmentation of the HGCAL enables the extraction of additional direction and shape observables.

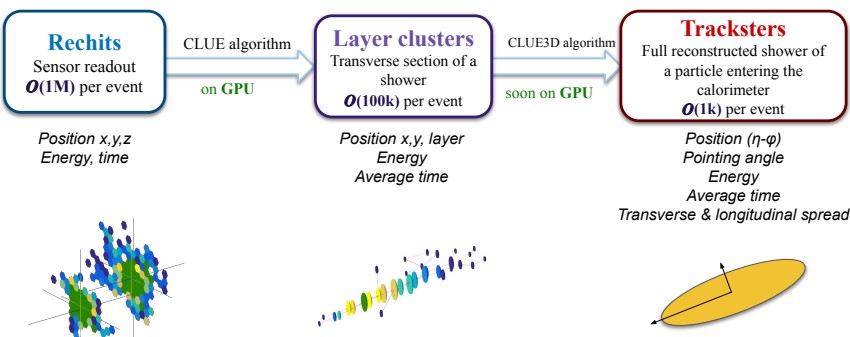

Figure 3: HGCAL reconstruction chain in the TICL framework, showing the different algorithms and their outputs. The multiplicity of each kind of object is shown for a typical 200 pileup event.

## 3    Particle identification

Particle identification (PID) algorithms are used to separate tracksters arising from hadrons from those arising from electrons and photons. The separation between hadronic and electromagnetic objects is crucial for controlling the computational time of electron reconstruction, since electromagnetic tracksters are used to seed the electron track reconstruction algorithms. The time requirements are more stringent at the High Level Trigger than in offline reconstruction, thus different strategies, with different trade-offs, have been investigated. Classical feature-based methods, based on high level observables such as the fraction of energy deposited in the hadronic compartment and the longitudinal/transverse shower spread, provide a limited PID capability but have a very fast computational time. Deep learning based algorithms can achieve significant performance improvements, however given the high multiplicity of objects in HGCAL, the inference time has to be kept under control. Layer clusters are used as input to a Convolutional Neural Network (CNN) or Graph Neural Network (GNN) [8], to exploit information on shower development. This approach balances input complexity ($\mathcal{O}(100)$ inputs per trackster) and classification performance, and is already demonstrating clear gains in separation power. This method is already in use in TICL. The most detailed approach uses all reconstructed hits ($\mathcal{O}(10^4)$ inputs per trackster), allowing the network to learn directly the granular shower topology. While computationally intensive, this method can achieve the best performance, particularly in dense pileup conditions.

## 4    Electron reconstruction

Electrons produced in proton-proton collisions radiate bremsstrahlung photons while traversing the CMS tracker, leading to spread-out energy deposits in the calorimeter. To recover these photons and maintain good electron momentum resolution, CMS employs *superclustering* [9], which combines calorimeter clusters associated with the parent electron and seeds track reconstruction with the Gaussian Sum Filter algorithm [10].

In the current detector, superclustering is performed with the *Moustache* algorithm [9], a geometrical method that collects clusters within a narrow window consistent with electron and photon patterns in the CMS detector. While effective under present conditions, this approach will face limitations in the HL-LHC environment, where pileup levels may reach 200 interactions per bunch crossing.

For HGCAL, a new superclustering strategy has been developed using a Deep Neural Net-

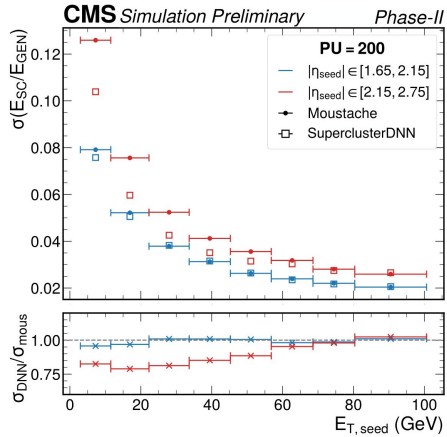

Figure 4: Energy resolution of superclusters, comparing the DNN-based and classical (Moustache) algorithm. [11]

work (DNN) trained on trackster features [11]. The model exploits kinematic, angular, and shape variables derived from HGCAL's fine granularity to distinguish genuine bremsstrahlung photons from pileup-induced clusters. Trained and benchmarked on simulated electrons with high pileup, the algorithm achieves a significantly better energy resolution than the Moustache algorithm at low energies and in the forward region, where pileup contamination is largest, as seen in Figure 4. At higher energies, the gain is marginal since pileup contributes predominantly at low energies.

## 5  Hadron energy regression

For hadronic tracksters, the energy resolution of the calorimeter can be considerably improved with machine-learning based regression. Using a Graph Neural Network, fed with all the reconstructed hits of a test beam setup with charged pions (without pileup), it was shown that an improvement in the energy resolution of hadronic showers up to a factor 2 was possible [12], as the neural network can learn the structure of the shower and partly compensate for energy leakage.

## 6  Conclusion

The High Granularity Calorimeter of CMS will be a key component of the High-Luminosity phase of the LHC, enabling precise energy, position, and timing measurements under the challenging pileup and radiation conditions. Its unprecedented granularity calls for a new reconstruction paradigm, implemented in the TICL framework and making large use of machine learning.

We have discussed several examples of such developments: particle identification with CNNs and GNNs, neural network-based superclustering for electron reconstruction, and energy regression of hadronic showers using a GNN. These approaches exploit the fine spatial granularity of HGCAL, achieving significant gains in performance compared to traditional methods, while keeping the needed computational power under control.

The results obtained so far demonstrate that machine-learning-based reconstruction can help meet the challenges of the HL-LHC, helping both offline reconstruction and event triggering.

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
