# Peer review of "Machine Learning for Event Reconstruction in the CMS Phase-2 High Granularity Calorimeter Endcap"

_SciPost Physics Proceedings_

## Round 1 · Referee Report · Anonymous (Referee 1) · 2025-11-27

Strengths

1- The paper is very clear. 2- It does a good job in giving a good overview of what the contribution was about in a limited space.

Weaknesses

Because of the space constraints

Report

I and a colleague have reviewed the paper named "Machine Learning for Event Reconstruction in the CMS Phase-2 High Granularity Calorimeter Endcap" with great interest. Despite the limited space, the paper achieves to give a good overview of the subject. I do not have any major comment on the content, just a few recommendations to improve the clarity of the paper.

Content:

  • At the end of section 1, it is mentioned that the HGCAL will feature 6m channels, and Figure 3 is suggesting that about 1M sensor will be active in a typical event. Taking the numbers at face value, this seems to imply an average occupancy of 15-20% which seems high. I believe this deserves a comment somewhere.
  • The usefulness of figure 4 is unclear. It seems a pretty standard picture of an electron doing bremsstrahlung. Perhaps the author can decide whether there is added value in the HGCAL context for figure 4, and, if so, clarify in the text.
  • It might be useful to better quantify the benefits of the ML approaches, for example by reporting numerical improvements observed in the cited reference (e.g., percentage gain in resolution, efficiency, background rejection) and/or by explicitly stating the metrics used to evaluate performance (AUC, ROC, etc.)

Minor editorial - Figure 2 is referenced before figure 1: how about swapping them? - I don’t see a reason to keep writing tricksters in italics after they have been introduced in the first paragraph of section 2.

Recommendation

Publish (easily meets expectations and criteria for this Journal; among top 50%)

  • validity: -
  • significance: -
  • originality: -
  • clarity: -
  • formatting: -
  • grammar: -

Author:  Théo Cuisset  on 2025-12-09  [id 6126]

(in reply to Report 1 on 2025-11-27)
Category:
answer to question

Dear referee,

Thank you for reviewing the paper. I have applied your remarks. You may find the answers inline between your comments below :

Content:

At the end of section 1, it is mentioned that the HGCAL will feature 6m channels, and Figure 3 is suggesting that about 1M sensor will be active in a typical event. Taking the numbers at face value, this seems to imply an average occupancy of 15-20% which seems high. I believe this deserves a comment somewhere.

The occupancy in HGCAL in a 200 pileup event is high, as the density of sensors is much higher in the regions of high activity (high pseudorapidity and in EM calorimeter). Occupancy is ranging from 5% to 60% in the silicon sensors depending on the region (the scintillators have low occupancy but represent only about 6% of the number of channels). (NB: These numbers are from the HGCAL TDR, so they are not completely up to date) I have added to the end of a sentence in the introduction (after "energy deposition and timing"): "In total, the calorimeter will feature about 6 million sensors, each capable of measuring both energy deposition and timing, with occupancy ranging from 60\% in the front of the EM section down to less than 1\% in the rear scintillator sensors~\cite{HGCALTDR}."

The usefulness of figure 4 is unclear. It seems a pretty standard picture of an electron doing bremsstrahlung. Perhaps the author can decide whether there is added value in the HGCAL context for figure 4, and, if so, clarify in the text.

The figure was purely for illustration. I removed it.

It might be useful to better quantify the benefits of the ML approaches, for example by reporting numerical improvements observed in the cited reference (e.g., percentage gain in resolution, efficiency, background rejection) and/or by explicitly stating the metrics used to evaluate performance (AUC, ROC, etc.)

In Section 5 (hadron regression), I have added a quantified improvement. The sentence now reads : "Using a Graph Neural Network, fed with all the reconstructed hits of a test beam setup with charged pions (without pileup), it was shown that an improvement in the energy resolution of hadronic showers up to a factor 2 was possible [12], as the neural network can learn the structure of the shower and partly compensate for energy leakage."

Minor editorial

Figure 2 is referenced before figure 1: how about swapping them?

done

I don’t see a reason to keep writing tracksters in italics after they have been introduced in the first paragraph of section 2.

done

---

## Round 2 · Author Response

Applied comments of referee.

---

## Round 2 · List of Changes

• added mention of the occupancy in the detector (along with citation of HGCAL TDR) in Introduction

  • removed Figure 4 (schematic description of a bremsstrahlung)

  • added a quantified improvement for energy resolution for hadron energy regression (Section 5)

---

## Editorial Decision

accepted_in_target_journal